# Cellular Dynamics of Fas-Associated Death Domain in the Regulation of Cancer and Inflammation

**DOI:** 10.3390/ijms25063228

**Published:** 2024-03-12

**Authors:** Kishu Ranjan, Chandramani Pathak

**Affiliations:** 1Department of Pathology, School of Medicine, Yale University, New Haven, CT 06511, USA; 2MGM School of Biomedical Sciences, MGM Institute of Health Sciences, Navi Mumbai 410209, Maharashtra, India; cmpathak@mgmuhs.com

**Keywords:** cancer, FADD, apoptosis, RIP kinases, autophagy, NF-κB, inflammation, therapy

## Abstract

Fas-associated death domain (FADD) is an adaptor protein that predominantly transduces the apoptosis signal from the death receptor (DR) to activate caspases, leading to the initiation of apoptotic signaling and the coordinated removal of damaged, infected, or unwanted cells. In addition to its apoptotic functions, FADD is involved in signaling pathways related to autophagy, cell proliferation, necroptosis, and cellular senescence, indicating its versatile role in cell survival and proliferation. The subcellular localization and intracellular expression of FADD play a crucial role in determining its functional outcomes, thereby highlighting the importance of spatiotemporal mechanisms and regulation. Furthermore, FADD has emerged as a key regulator of inflammatory signaling, contributing to immune responses and cellular homeostasis. This review provides a comprehensive summary and analysis of the cellular dynamics of FADD in regulating programmed cell death and inflammation through distinct molecular mechanisms associated with various signaling pathways.

## 1. Introduction

Fas-associated death domain (FADD), also known as MORT-1, is an adaptor protein within the TNF receptor family, enabling molecular interactions between death receptors (DRs) and apical procaspase-8 and -10. This interaction leads to the formation of a multimeric death-inducing signaling complex (DISC) crucial for initiating apoptosis signaling [1,2]. FADD, also recognized as a multifunctional adaptor protein, has been identified to participate in a wide range of cellular signaling processes, including cell proliferation, autophagy, necroptosis, inflammation, embryogenesis, and innate immunity [1,3]. FADD has emerged as a crucial player in the regulation of cancer and inflammatory diseases [4]. Upon the binding of a ligand to the cell surface DRs, the death domain (DD) of the DRs engages in homophilic interactions with the DD of FADD. This interaction subsequently triggers the oligomerization of the death effector domains (DEDs) of FADD with apical caspases, namely procaspase-8/-10, leading to the formation of a death-inducing signaling complex (DISC) [1]. Since the predominant function attributed to FADD was understood to involve the transmission of DR signaling by interacting with membrane-bound death receptors, it was initially presumed that FADD was predominantly located in the cytoplasm. However, subsequent observations identified its localization in the nucleus as well. This discovery suggests that the expression of FADD in the nucleus may confer cellular protection against apoptosis [5,6]. The exact molecular mechanisms responsible for cell survival through nuclear FADD have yet to be determined. Importantly, the mechanistic significance of FADD in cytoplasmic to nuclear trafficking is mostly associated with distinct cell and cancer types [7]. Constitutive expression of FADD aids cellular homeostasis; however, defective or low expression of FADD has been implicated in pathological manifestations in different types of cancers [8]. In fact, mechanistically evasion of apoptosis through low expression of FADD and elevated expression of cFLIP attributes inhibition of DR-mediated apoptosis and favors developing drug resistance in cancer cells [9,10]. Although the expression of FADD is a contentious topic in various cancer types, the precise molecular mechanisms remain incompletely understood. Previous research has suggested that the apoptotic effects of FADD, whether pro- or anti-, are contingent upon the particular tissue contexts [11]. It is well accepted that FADD, as an adaptor protein, provides the binding platform for many cell death regulatory proteins—caspase-8/-10, cFLIP, and RIP kinase—to define the fate of cell death or survival. It could be hypothesized that post-transcriptional modifications, mutations in the FADD gene, changes in phosphorylation, or mutations in death receptor (DR) genes in various tissue types may lead to modifications in FADD expression. Investigating the role of FADD in developmental processes and cancer biology is a compelling area of research. FADD has a parallel role in the regulation of cell proliferation and programmed cell death. The involvement of FADD has been shown to regulate T and B cell development in immune response [7]. Moreover, studies have demonstrated that the phosphorylation of FADD may play an additional role in the regulation of cell proliferation, although the kinases responsible for the altered phosphorylation of FADD remain poorly understood [8,12]. Previous studies have demonstrated the phosphorylation of FADD and its impact on anticancer drug efficacy in human prostate cancer cells. These studies have proposed that the phosphorylation of FADD at Ser194 influences both the upstream and downstream components of the MEKK1/MKK7/JNK1 signaling pathway, as well as the responsiveness of anticancer drugs [12]. Phosphorylation of FADD is linked to the regulation of the cell cycle and cell proliferation [13,14]. An earlier report showed that a deficiency of FADD leads to embryonic lethality and impaired T cell proliferation [15]. Indeed, deficiency of FADD also leads to dysregulation of - cell cycle progression [16]. Notably, FADD is also attributed to autophagy signaling by interacting with DD of Atg5 to promote type II programmed cell death signaling [17]. Moreover, deficiency of FADD in T cells triggers autophagic signaling and renders the cells susceptible to caspase-independent cell death. The multifaceted role of FADD has also been documented in RIP1- and RIP3-dependent necroptosis signaling [18,19].

Various signal transduction factors and cellular mechanisms are intriguingly involved in the regulation of cell death and inflammation through FADD. These include pro-caspase-8, cFLIP, RIPK1, RIPK3, E3 ubiquitin ligase TRAF2, the deubiquitinating enzyme A20, and the inhibitors of apoptosis (IAPs), which play a crucial role in determining whether cells undergo programmed cell death or activate survival pathways [20]. Receptor-interacting protein kinase (RIPK1) and RIPK3 are key mediators of necroptosis and have a vital role in the regulation of cell death, independent of proinflammatory signaling. Apoptosis and necroptosis are intricately governed biological processes that are orchestrated by specialized proteins, namely RIPK1, RIPK3, and mixed lineage kinase domain-like (MLKL), in conjunction with FADD [4]. MLKL stands out as a vital effector in the mechanism of necroptosis. Research has shown that both MLKL and FADD play pivotal roles in mitigating lymphoproliferative disorders and modulating the activation of the NLRP3 inflammasome [21]. A recent study has shown that the activation of the inflammasome can lead to apoptosis and pyroptosis, followed by the activation of caspase-1 under specific conditions [22]. The activation of the inflammasome initiates the oligomerization of the adaptor protein apoptosis-associated speck-like protein containing a caspase recruitment domain (ASC) and caspase-1/8. This process regulates the maturation of cytokines (IL-1β and IL-18), influencing inflammation or cell death [23]. Caspase-1-mediated inflammasomes trigger the cleavage of gasdermin D (GSDMD) to induce programmed necrosis (pyroptosis) [24]. Notably, caspase-8 may also act as a regulator of GSDMD-driven pyroptotic cell death [25,26]. The exact mechanism through which caspase-8 triggers GSDMD activation is currently not fully understood. The involvement of FADD, cFLIP, caspase-1/-8 in programmed pyroptosis, apoptosis, and necroptosis represents a complex puzzle within the realms of inflammation and cancer. Many types of cancer exhibit proliferative inflammatory atrophy in adjacent tissues, a condition resulting from inflammation and cellular damage that leads to the upregulation of the cell death regulatory protein Bcl-2 [27]. The increased expression of Bcl-2 has been established to inhibit apoptosis and enhance cell viability in cancer. Moreover, Bcl-2 also modulates pyroptosis and necroptosis through its interaction with BH3-like domains in GSDMD and MLKL [28]. The activation of the inflammasome is also associated with the regulation of the mammalian target of rapamycin (mTOR), a serine/threonine kinase that plays a role in the differential regulation of pro- and anti-inflammatory cytokine levels [29]. Thus, FADD may serve as a crucial scaffold in regulating inflammatory responses, in addition to playing a key role in various cell death pathways that influence cellular outcomes.

In the canonical pathway, DR stimulation in the presence of FADD proteolytically activates caspase-8 to cleave RIPK1 and RIPK3, thereby preventing necrosis [30,31]. Interestingly, FADD plays an important role in inflammatory signaling and related disorders. Previous studies have demonstrated the importance of FADD in the regulation of NF-κB and toll-like receptor (TLR) signaling in modulating interferon response against infectious exposures [4,32]. Moreover, TNFα stimulation induces linear ubiquitination of endogenous FADD in Jurkat cells to induce pro-survival mechanisms [33]. In contrast, the E3 ligase Makorin Ring Finger Protein 1 (MKRN1) mediates ubiquitination and proteasomal degradation of FADD, thereby abrogating the activation of cell death. It has been shown that MKRN1 knockdown results in FADD protein stabilization and the formation of the DISC, which causes hypersensitivity to extrinsic apoptosis [34]. We have previously reported that FADD induces JNK1-dependent ubiquitination of the cFLIP protein to instigate death receptor-mediated apoptosis [35,36]. Importantly, further thorough investigation of FADD-mediated ubiquitin signaling is warranted, and identifying new regulators of FADD holds greater promise for the development of targeted therapies. Overall, FADD has multiple regulatory functions and serves as a unique molecule to regulate both apoptotic and non-apoptotic signaling. This review article offers a comprehensive understanding of the underlying molecular mechanisms of cellular dynamics of FADD, which serves as a pivotal signaling component regulating various cellular functions to maintain cellular integrity and determine cell fate.

## 2. Structure and Cellular Localization of Fas-Associated Death Domain (FADD) 

### 2.1. The Structural Organization of FADD 

The human FADD gene is located on chromosome 11q13.3 and consists of two coding exons that encode a 22 kDa protein [37]. FADD contains two functional domains: an N-terminal Death Domain (DD) and a C-terminal Death Effector Domain (DED) (Figure 1A). The DD of FADD (FADD-DD) consists of 80 amino acids arranged in six antiparallel amphipathic α-helices, which structurally resemble the CD95 DD. The FADD-DD is crucial for homophilic interaction with the DD of death receptors [38,39,40]. FADD-DD interacts with multiple receptors, including TRADD (TNFR-1 signaling), DR3, TRAIL receptors 1 (DR4), and TRAIL receptors 2 (DR5), to activate extrinsic apoptosis signaling [41,42]. The FADD-DED interacts with DEDs of procaspase-8 and/or -10, forming a death-inducing signaling complex (DISC), which facilitates downstream apoptosis signaling [43,44] (Figure 1A). It is worth noting that the orthologs of human FADD have been characterized in mice and Xenopus. The mouse FADD (mFADD) and Xenopus FADD (xFADD) proteins have 80% and 62% structural resemblance with human FADD (hFADD), respectively [45,46]. Induced expression of xFADD in mammalian cells leads to apoptosis, while a truncated (dominant negative) DED of xFADD abolishes apoptosis in response to Fas ligand stimulation [46]. The structural analysis of FADD has revealed that the DD and DED domains of FADD are arranged in an orthogonal tail-to-tail manner, with each domain having a conserved backbone of six α-helices [37,47]. Further, studies have shown that the DD of FADD is enriched in positively charged residues (K110, R113, R114, R117, R127) near the α11/α12 interhelical loop, allowing for interaction with the DD of CD95/Fas receptor [47,48]. Moreover, the interfaces of the first and sixth α-helices of FADD-DD heterodimerize with DD-containing proteins involved in cell death and inflammatory signaling [39,49]. Previous in vitro studies have demonstrated that purified DD of FADD can interact independently with the DD of receptor-interacting protein kinase 1 (RIPK1 or RIP1) and play a role in regulating the necrotic activities of RIPK1 [50]. Notably, NMR structural analysis has shown that each α-helix of FADD-DED is rich in conserved hydrophobic and negatively charged residues, such as Glu/Asp or Asn (at position 19) and the R_X_DL motif (at positions 78–81).

This provides a platform for DED-containing proteins to assemble DISC [51,52], although the ectopic expression of FADD in mammalian cells induces apoptosis. However, truncated (dominant negative) DED of FADD abrogates apoptosis in response to Fas ligand stimulation [46]. Thus, structural analysis of FADD revealed that individual domains of this protein are crucial for normal cell proliferation and regulation of the cell cycle to proper development apart from its involvement in apoptotic signaling (Figure 1A).

### 2.2. Localization, Phosphorylation, and Expression of FADD Governs Its Function and Cellular Fate

The cellular localization, phosphorylation, and expression of FADD play a significant role in determining cellular function by anticipating DD or DED through interactions with various proteins containing DD or DED domains. The cellular localization, expression levels, phosphorylation status, and interactions with binding partners dictate the activation or inactivation of downstream signaling pathways, thereby determining its function and influencing the cell’s fate regarding survival or cell death [1]. The expression level of FADD can be regulated by various factors, such as intrinsic or extrinsic stimuli, cell type, developmental stage, and stress signals. The cytoplasmic and nuclear expression of FADD has been observed in both human and mouse cell lines in vitro [53]. It has been postulated that FADD primarily resides in the nucleus and is subsequently shuttled to the cytoplasm [53,54]. The cytoplasmic presence of FADD induces apoptotic cell death, while its nuclear localization may function to protect the cell from apoptosis [1,9]. Moreover, an intriguing study demonstrated that activation of CD95 leads to the redistribution of FADD from the nucleus to the cytoplasm [55]. Thus, these findings suggest that the nuclear localization of FADD protects cells from apoptosis, while its cytoplasmic localization promotes cell death (Figure 1B). The cytosolic presence of FADD plays a critical role in death receptor-induced death-inducing signaling complex (DISC) formation and apoptosis signaling [1]. Coupled with the observation that FADD plays a dual role in the regulation of programmed cell death, the expression of FADD is pivotal in determining cell survival and immune responses. Conversely, dysregulation of FADD expression leads to a range of pathological outcomes, such as cancer, autoimmune disorders, type 2 diabetes, and inflammatory diseases [9,56]. Previous research, including our study, showed that overexpression of full-length FADD can induce apoptosis either independently or in response to death receptor activation [35,36,38,57]. Mutagenesis and biochemical studies have further revealed that FADD-DD alone cannot initiate downstream apoptosis signaling without the activation of the death receptor and requires functional DEDs. Interestingly, over-expression of DED is sufficient to induce cell death without the need of death receptor [47]. In contrast, overexpression of FADD-DD inhibits downstream signaling and activation of Fas/CD95 and DR5 receptors mediated apoptosis [45,58]. Mutational analysis has shown that deletion or mutation in the DED of the FADD (FADD-DN) acts as a dominant negative in death receptor signaling and loss the ability to recruit executioner caspase-8 [59]. The DED mutation of FADD (deletion of 80-208) has been demonstrated to result in impaired functionality of FADD in response to canonical death inducers and defects in T cell proliferation [60]. These studies suggest that mutations in the FADD gene impair functional outcomes and may also be responsible for alterations in gene expression, phosphorylation, or ubiquitination.

Post-transcriptional modifications (PTMs) constitute a dynamic process aimed at modulating the functionality, subcellular localization, and stability of proteins. Similarly, the FADD protein undergoes PTMs, such as phosphorylation, ubiquitination, and SUMOylation [61]. Currently, the functional implications of FADD-driven cellular signaling also depend on its phosphorylation status. FADD has been identified as a substrate for various kinases, such as FIST/HIPK3 kinase, casein kinase (CK) Iα, Aurora-A kinase (AURA), and polo-like kinase (PLK)1. Specifically, FIST/HIPK3 (Fas/FADD-interacting serine/threonine kinase) has been observed to induce the phosphorylation of FADD, leading to the inhibition of fas-mediated Jun NH(2)-terminal kinase (JNK) activation [62]. A crucial role of FADD in non-apoptotic functions related to cell cycle progression and proliferation was unveiled through the phosphorylation of serine 194 in the C-terminus of FADD by casein kinase Iα (CKIα) (Figure 1A,B). Furthermore, phosphorylation of FADD at Ser200 indicated its nuclear localization, which is influenced by Casein kinase 2 (CK2). Elevated level of CK2 have been detected in malignant cells and various types of cancers, such as lung, colon, breast, ovarian, pancreatic, and prostate cancer [13,63]. Interestingly, phosphorylation of FADD at serine 194 mediates the nuclear localization of FADD for genomic surveillance [54]. Moreover, CK1α induces the phosphorylation and nuclear translocation of FADD, while CK2β retains phosphorylated FADD inside the nucleus to inhibit death receptor signaling [54,64]. The shuttling of FADD between the cytoplasm and the nucleus occurs through Exportin-5 (Figure 1B). This nucleo-cytoplasmic shuttling protein interacts with FADD to support the translocation of phosphorylated FADD (pFADD) into the nucleus. A mutation at Serine 194 (Ser194) residue of FADD impairs interactions with Exportin-5 [8,54]. The nuclear translocation of pFADD strengthens the anti-apoptotic activity of NF-κB, promoting cell proliferation [14]. A previous report highlights that FADD predominantly translocate to the high non-condensed transcriptionally active region of chromatin, but in the absence of DNA binding motifs, FADD may not directly influence the transcriptional machinery [65]. Nevertheless, the DED of FADD interacts with the methyl-CpG binding domain protein 4 (MBD4) in the nucleus; however, the downstream signaling of FADD-MBD4 is not well defined [54]. Cytosolic FADD interacts with the death receptor and autophagy intermediates [66], while nuclear localization of FADD strengthens its anti-apoptotic activity and promotes cell proliferation [14]. Subsequently, an additional study demonstrated that the nuclear localization of FADD relies on strong nuclear localization signal (NLS) and nuclear export signals (NES) in the DED, which facilitate the nucleo-cytoplasmic shuttling of FADD [67]. Mutation in a phenylalanine residue at position 25 of the DED abolishes the nuclear translocation of FADD [54]. Another interesting report revealed that phosphorylation of FADD at Ser203 in response to Taxol by Aur-A and Plk1 triggers both apoptotic and necrotic cell death [68]. In addition, the G2/M stage has been identified as the most favorable phase for FADD phosphorylation during the cell cycle process, although the molecular mechanism behind these remains unclear [69]. Remarkable evidence from structural analysis of FADD underscores the crucial role played by the individual DD and DED domains in transmitting cell death signaling. Furthermore, the phosphorylation state of FADD regulates the inhibition of caspase-8 activation and Fas resistance, thereby safeguarding cells from apoptotic death. Moreover, research has demonstrated that FADD interacts with protein kinase C (PKC)ζ. The ensuing phosphorylation of FADD leads to the protection of Fas-mediated cell death and DISC formation [70]. In recent studies, multiple E3 ubiquitin ligases, such as Makorin Ring Finger Protein 1 (MKRN1), C terminus HSC70-Interacting Protein (CHIP), and Linear Ubiquitin Chain Assembly Complex (LUBAC), have been implicated in the degradation of FADD [33,34]. Moreover, the mitochondrial E3 ubiquitin ligase (MAPL) has been identified as a mitochondrial SUMO E3 ligase. In this context, the RING (really interesting new gene) domain of MAPL is exposed to the cytoplasm and is responsible for SUMOylating dynamin-related protein (DRP1) [71]. SUMOylation of FADD enhances DRP-1 interaction to facilitate mitochondrial fragmentation during apoptosis and necrosis [72]. Collectively, these findings suggest that FADD is involved in regulating various cellular processes beyond cell death to maintain cellular homeostasis.

Next, the expression of FADD is crucial for mediating both cell death and survival functions. Varied tissue-specific expression patterns of FADD have been observed in healthy human tissues. Dysregulated FADD expression, including overexpression and downregulation, has been documented in various types of cancer [11]. In general, a decreased expression of FADD was noted in conditions such as acute myeloid leukemia, thymic lymphoma, glioblastoma, pancreatic cancer, colorectal cancer, renal cancer, and prostate cancer. Conversely, an increased expression of FADD was observed in lung cancer, head and neck carcinoma, breast cancer, liver cancer, and urothelial cancer [11]. The existing literature has unveiled that the expression and prognostic significance of FADD in the context of cancer remains enigmatic, primarily attributable to its pleiotropic effects and dual roles in apoptosis, as well as its involvement in intricate cellular signaling pathways. Numerous transcription factors (TFs) are recognized for their role in regulating the expression of FADD in cancer cells (Figure 2; detailed analysis in Appendix A). The TF, hypoxia-inducible factor-1α (HIF-1α) inhibited the transcriptional activity of the FADD gene in colon cancer cells [73]. Moreover, previous studies have demonstrated that the overexpression of BRCA1 in breast cancer cells that lack BRCA1 gene significantly upregulates FADD expression. This increase can be attributed to the direct interaction between BRCA1 and the promoter region of FADD. Conversely, the depletion of BRCA1 has been shown to cause a marked decrease in FADD expression, both at the protein and messenger RNA (mRNA) levels [74].

## 3. Molecular Interaction of Fas-Associated Death Domain and Functional Tale

### 3.1. FADD and cFLIP Interactions

Cellular FLICE (FADD-like IL-1β-converting enzyme)-inhibitory protein (cFLIP) plays a major role in blocking death receptor (DR) mediated apoptosis. It also plays a fundamental role in apoptosis, immune receptor signaling, inflammation, autophagy, and necroptosis [10,76]. Similar to procaspase-8/-10, the anti-apoptotic protein cFLIP also contains two death effector domains (DEDs) at the N-terminal, but it is an inactive enzymatic homolog of procaspase-8/-10 [77] (Figure 3A). The gene encoding regions of *cFLIP* are located near *procaspase-8* and *procaspase-10* on chromosome 2q33, suggesting a genetic link between these apoptosis regulatory genes [78]. The cFLIP protein has three isoforms: a long form cFLIP_L_ (55 kDa), a short variant cFLIP_S_ (27 kDa), and regulator cFLIP_R_ (25 kDa). All isoforms of cFLIP contain two death effector domains (DEDs), but cFLIP_L_ has an additional inactive caspase-like DED, while cFLIP_S_ and cFLIP_R_ lack the entire caspase-like DEDs [76,79] (Figure 3A). Importantly, all three isoforms of cFLIP are believed to competitively inhibit procaspase-8 recruitment to the DISC through their DEDs [80,81]. In cancer cells, cFLIP occupies the majority of FADD, which serves as a common docking site for both procaspase-8 and cFLIP through DED interactions [35,36,82]. Computational analysis suggests that the DEDs of FADD, cFLIP, and procaspase-8 contain a ‘charged triad’ E/D-RxDL motif that is crucial for their downstream signaling [49]. Molecular docking studies have shown that FADD DED preferentially engages FLIP through its α1/α4 surface and procaspase-8 using its α2/α5 surface. These relative orientations contribute to FLIP being recruited to the DISC at comparable levels to procaspase-8 despite lower cellular expression [83]. Additionally, cFLIP has a higher binding affinity to FADD compared to procaspase-8 at the DISC [84]. The heterogeneous expression of FADD and cFLIP across different tumor types confers resistance to death receptor-induced apoptosis (Figure 4). Mechanistically, in response to TNF-α, TNF receptor (TNFR1) oligomerize with adaptor proteins TRADD, ubiquitin ligases TRAF2 and cIAP1/2, and RIP1 to form ‘complex I’ for the activation of NF-κB signaling [85]. At the transcriptional level, activated NF-κB induces the expression of anti-apoptotic proteins such as cFLIP, cIAPs, and Bcl-2 family members to block apoptotic signaling [86,87]. Importantly, elevated expression of cFLIPL strengthens the complex I for constitutive NF-κB activation [88,89]. Notably, the cFLIP protein undergoes post-translational modifications [90,91]. The phosphorylation of cFLIPL and cFLIPS has been observed in various cell lines, inhibiting their interaction with the adaptor molecule FADD and sensitizing cancer cells to apoptosis [92]. Furthermore, ubiquitin-mediated protein modification of cFLIP isoforms is essential for cellular homeostasis and proliferation [93,94]. We have previously shown that TNF-α stimulation of FADD-overexpressing cells induces JNK and E3 ligase ITCH-dependent ubiquitination of cFLIPL [35]. Panner et al. demonstrated the PTEN-Akt-AIP4-mediated ubiquitination process of FLIP_S_, resulting in enhanced sensitivity to TRAIL in glioblastoma multiforme [95]. Additionally, it was found that the lysine 167 residue (K167) of cFLIP_L_ functions as a newly identified site for ubiquitination, leading to ROS-dependent degradation [96]. Furthermore, the DNA repair protein Ku70 was discovered to interact with cFLIP, thereby shielding it from polyubiquitination and subsequent proteasomal degradation [97]. However, despite significant advances in understanding the regulation of apoptosis and cell survival, the involvement of FADD and cFLIP in these processes remains elusive.

### 3.2. FADD and RIP1 Interaction

RIP1 is an adaptor protein containing a death domain (DD) and is a crucial component of TNF-R1 and TRAIL-R1/R2 signaling [85,99] (Figure 3B). The importance of the receptor interacting protein kinase 1 (RIPK1) in cellular homeostasis was investigated in *RIP1*-deficient mice, which exhibited lethality due to extensive apoptosis in both lymphoid and adipose tissue [100]. It was observed that an interplay between FADD and RIP1 is critical for the regulation of apoptosis and necrosis during embryogenesis and lymphocyte function in a mouse model [101]. The heterogeneous expression of FADD and RIPK1 exacerbates the restriction of interaction and subsequent signaling in diverse tumor types. (Figure 4). Additionally, *Rip1* kinase inactive mutations have distinct impacts on the embryogenesis of *Fadd*-deficient mice [102]. The knockdown of *FADD* or *FADD*^−/−^ leukemia Jurkat T-cells failed to induce NF-κB signaling upon TRAIL stimulation [103]. Upon ligand-dependent receptor activation, intracellular FADD oligomerizes and recruits RIP1 and caspase-8, forming a complex called the RIPoptosome, which initiates programmed necrosis (necroptosis) [104]. Jang et al. demonstrated that the RIP1 DD and FADD DD form a stable complex with a structure similar to that of the Fas DD/FADD DD complex [105]. Earlier structure-based mutagenesis studies revealed that RIP1 DD point mutations K604E, E614K, G623K, E626K, M637K, K642D, and S657K disrupt the stability of the complex with FADD DD, suggesting that the RIPoptosome and the Fas death-inducing signaling complex share a common assembly mechanism (Figure 3B) [105]. The pleiotropic nature of TNFR1 signaling plays an essential role in regulating apoptotic and non-apoptotic signaling pathways. Activation of TNFR1 leads to the formation of ‘complex I’, which includes TNF receptor-associated protein with a death domain (TRADD), TNF receptor-associated protein 2 (TRAF2), RIP1, and cellular inhibitor of apoptosis proteins 1 and 2 (cIAP1 and cIAP2), and primarily regulates NF-κB signaling [85]. Moreover, during the regulatory inhibition of NF-κB signaling, TNFR1 signaling recruits FADD, caspase-8, and RIP1 to form a cell death-inducing complex known as ‘complex II’ [104,106]. Interestingly, a previous report demonstrated that RIP1-deficient cells failed to induce NF-κB activation, even in the absence of TNF-α stimulation [100]. Notably, NF-κB activation can be blocked by TNF-α-mediated signaling through caspase-8 mediated cleavage of RIP1 [104,107]. These studies highlight the pleiotropic role of RIP1 in maintaining cellular homeostasis by regulating apoptosis machinery and cell survival pathways. Stimulation of TNFα leads to the autophosphorylation and polyubiquitination of RIP1. Polyubiquitination through Lysine 48 (K48) linkage leads to degradation, while Lysine 63 (K63) linkages result in the activation of IκB kinase and NF-κB activation [106]. Furthermore, a point mutation of RIP1 at Lysine 377 (K377R) blocks K63-linked polyubiquitination, preventing the recruitment of IκB kinase, IKKβ, and TAK1 complex to the TNF receptor and thereby inhibiting NF-κB activation [108]. Therefore, polyubiquitination of RIP1 is crucial for the activation of IKKβ, which phosphorylates IκB, an inhibitor of NF-κB, leading to its degradation via the proteasomal pathway [109]. As a result, NF-κB is released from the inhibitory complex, translocated to the nucleus, and activates the transcription of target genes involved in immunity, inflammation, and survival [110]. Downregulation of RIPK1 in HepG2 cells significantly reduces NF-κB transcriptional activity and promotes caspase-8 and caspase-3-mediated apoptosis [111]. In summary, in the absence or downregulation of complex I-mediated NF-κB activation, the RIP1-FADD-caspase-8 complex II reinforces apoptosis. In the following sections, we will discuss the details of RIPoptosome signaling.

## 4. FADD in Regulation of Programmed Cell Death and Inflammatory Signaling

### 4.1. FADD in the Regulation of the TNFα-NF-κB Signaling Axis

TNF-α-induced NF-κB signaling and downstream activation of anti-apoptotic genes have a negative impact on apoptosis signaling. In the majority of cancer types, abnormal activation of NF-κB signaling promotes tumor development [85,112]. In addition to cancer cell signaling, dysregulation of TNF receptor (TNFR) signaling is associated with inflammatory disorders, such as arthritis and inflammatory bowel disease [113,114,115], making it a promising therapeutic target. The role of FADD in regulating TNF-α induced NF-κB signaling activation has been the subject of ongoing debates, with several groups currently working to determine the underlying mechanisms. TNF-α is a multifunctional cytokine belonging to the tumor necrosis factor superfamily, with important roles in cellular immunity, cell differentiation, proliferation, inflammation, and cell death [86]. Dysregulation of NF-κB signaling is closely associated with various human diseases, including cancer [116]. Activation of NF-κB-associated signaling for evading tumor cell death is a major factor contributing to tumor cell proliferation [87]. We and others have previously demonstrated that the TNFα-NF-κB signaling axis promotes prolonged survival in various tumor cell types [35,36], but these cells remain susceptible to apoptosis induction by chemotherapeutic drugs and radiation [117,118]. TNF-α exerts its biological effects through cell surface TNF receptors (TNFRs), which consist of a cytoplasmic death domain (DD) of approximately 80 amino acids in length. TNFR-DD is responsible for recruiting downstream components of the death machinery upon TNF-α stimulation [86]. Activation of TNFR-1 leads to a conformational change in its cytoplasmic DD tail, allowing it to interact with the DD-containing adaptor protein TRADD (TNFR-associated death domain). TRADD can form both a pro-inflammatory/survival “complex I”, which recruits RIP1, TNFR-associated factor (TRAF)-2 and -5, and cIAP 1/2, as well as a pro-apoptotic signaling “complex II”, which recruits FADD and RIP1 [99,104]. While the DD of RIP1 can directly interact with the DD of ligand-bound TNFR1, it generally prefers TRADD-mediated recruitment in complex I, possibly due to its high affinity for TRADD [119]. Formation of complex I lead to robust activation of NF-κB and AP-1 and upregulates several anti-apoptotic genes such as Bcl-xL, A1/Bfl-1, (c-IAP) 1/2, X-chromosome-linked IAP (XIAP; also known as hILP), and cFLIP [85,87]. Moreover, the binding of transforming growth factor-β-activated kinase (TAK-1) binding protein (TAB)-2/TAB-3 to ubiquitinated RIP1 stabilizes complex I, further activating NF-κB [120,121]. In contrast, the deubiquitinylation of RIP1 by the enzymes cylindromatosis (CYLD) or cIAPs subsequently dissociates RIP1 from complex I and allows I to interact with FADD and procaspase-8 forming complex II and triggering cell death [91,106]. Importantly, the oligomerization of the RIP1-FADD-procaspase-8 complex II is tightly regulated by the anti-apoptotic protein cFLIP [106]. In this context, tumor cell survival signaling pathways, including NF-κB, MAPK/ERK, and Akt, are known to transcriptionally upregulate cFLIP expression in a feedback mechanism [88]. We have demonstrated that the expression of FADD is critical for maintaining complex II-mediated apoptotic cell death by regulating the expression of cFLIP and the assembly of complex I [35] (Figure 3C). Zhou et al. showed that pharmacological targeting of IAPs suppressed NF-κB activation and induced FADD-dependent apoptosis in multiple myeloma (MM) cells, and cells expressing dominant-negative FADD were markedly less sensitive to IAPs inhibitors, highlighting the significant functional contribution of FADD in targeting anti-apoptotic NF-κB activation [122]. Chaudhary et al. previously demonstrated that low FADD concentration induces the activation of NF-κB signaling in a time- and dose-dependent manner [123]. Furthermore, bifurcated TNF-α signaling in the form of “complex I” and “complex II” represents independent mechanisms that may be specific to certain cell types [104,124]. The expeditiously formed complex I is composed of TNFR1, TRADD, RIP, TRAF2, and c-IAP1, inducing an NF-κB response without triggering apoptosis. On the contrary, complex II, lacking TNFR1 but including FADD, procaspase-8, and -10, initiates apoptosis when the NF-κB signal from complex I fail to induce the synthesis of antiapoptotic proteins like cFLIP_L_. Subsequent studies have revealed a previously unappreciated role for the FADD protein as a molecular switch regulating the TNF-α- NF-κB signaling axis, thereby influencing both apoptosis and cancer cell proliferation (Figure 3D).

### 4.2. Role of FADD in Necroptosis

Necroptosis is a form of caspase-independent cell death that is mediated by the RIPK3 protein. RIPK3 phosphorylates and activates the pseudokinase Mixed Lineage Kinase-Like (MLKL), which then executes cell death in the absence of apoptotic pressure [125]. However, some apoptotic components have been found to be involved in the formation of the necroptotic complex and dependent cell death [126,127,128]. The necroptosis signaling regulates various cellular and pathological processes, ranging from development to the regulation of immune regulatory cells [19]. Necroptotic cell death is initiated by the cytokines TNF-α, Fas, or TRAIL, which leads to dysregulation of mitochondrial reactive oxygen species (ROS) production and eventual collapse of cellular energy production [129,130]. Additionally, genotoxic agents and related cellular stress can also induce necroptosis [106,131].

The C-terminal DD of RIPK3 facilitates its interaction with the DD of RIPK1 through a RIP homotypic interaction motif (RHIM). The interaction leads to auto-phosphorylation and activation of RIPK3 as well as the assembly of “complex IIb”. At the molecular level, the RIPK1-RIPK3 complex IIb associates with FADD and caspase-8 to form the necroptotic complex [19,128,132]. It is worth noting that FADD and caspase-8 are common components in complex II and complex IIb, and they constantly regulate RIPK3-mediated necrotic cell death [132,133]. Previous studies have shown that the deletion of RIP3 completely restores cell proliferation in FADD-mutated T cells [134,135]. Conversely, FADD-deficient primary T cells fail to assemble the RIP1–RIP3 complex [136]. Furthermore, mouse embryonic fibroblasts (MEFs) lacking FADD show resistance to TNF-α-induced necrosis, and restoration of FADD expression restores both apoptotic and necrotic sensitivity to TNF-α [137]. Irrinki et al. demonstrate that the FADD-RIP1-RIP3-NEMO (NF-κB essential modulator) complex induces disintegration of mitochondrial bioenergetics to promote TNF-α-driven necroptosis [138]. An incoherent expression of FADD and RIPK2 is consistently observed across various cancer types, resulting in improper interaction and downstream signaling (Figure 4).

*Ripk1*-deficient mice can survive for a few days [100], and co-deletion of *fadd* or *caspase-8* does not prevent the perinatal lethality of ripk1^−/−^ mice [139]. However, the combined deletion of FADD-*caspase-8*-mediated apoptosis and RIPK3-MLKL-mediated necroptosis provides protection from lethality and increased survival [140]. In MEFs, the ablation of *Rip1* challenges TNF-α-induced expression of cFLIP, suggesting that TNF-induced caspase-8-mediated apoptosis partially contributes to the lethality of newborn *ripk1*^−/−^ mice [141]. Furthermore, FADD has been reported to suppress RIP3-mediated chronic intestinal inflammation and necrosis in epithelial cells [18]. Conversely, Fas signaling mediated suppression of RIP3 by caspase-8 or FADD facilitates T cell clonal expansion [66,135]. FADD has also been shown to neutralize virus-induced interferon production by enhancing the ubiquitin activity of the E3 ligase TRIM21 [142]. Importantly, the E3 ubiquitin ligase MKRN1 regulates the ubiquitination of FADD and RIP1–RIP3 complex formation [34]. Thus, FADD-mediated regulation of necroptosis signaling could provide an opportunity to define the fate of cells in pathological conditions.

### 4.3. Role of FADD in Autophagy

The process of autophagy, commonly referred to as the “self-eating” process, aims to degrade unwanted cytosolic constituents by transporting them to the lysosome to protect against stress-induced cell death [143]. While basal levels of autophagy help maintain cellular homeostasis, autophagy can induce cell death during pathological or physiological stress [144]. The pathways of apoptosis and autophagy are interconnected through key regulatory proteins that govern cell death and survival [145]. The DD of FADD interacts with autophagy-related protein 5 (ATG5), thereby triggering autophagic cell death in response to IFN-gamma stimulation [17]. Previous studies have reported that Atg5 has a dual role in the regulation of autophagy, but under cell death stress, it may induce cell death [17]. Pua et al. observed significantly increased cell death in Atg5^−/−^ CD8+ T lymphocytes and proposed that the co-regulation of Atg5 and FADD, either in the autophagic process or independent of autophagy, may transmit signals crucial for T cell proliferation [146]. Pyo et al. demonstrated that a lysine residue in the middle and C-terminal region of Atg5 is conjugated with Atg12 and binds to FADD to induce cell death [17]. Additionally, Pyo et al. used immunoprecipitation analysis to detect the association of Atg5 and Atg12 with FADD in a complex [17]. Earlier reports have indicated that FADD and caspase-8 jointly regulate autophagic signaling for proper T cell proliferation [146,147]. Mitogenically activated T cells trigger the interaction between FADD and the Atg5:Atg12 complex, leading to caspase-8 activation and autophagic cell death [146]. Depletion of FADD forces T cells to undergo hyperautophagy and activate RIP1-dependent necroptotic cell death, independent of caspase-8 activation [148]. Re-expression of full-length FADD in FADD^−/−^ MEF restores basal-level autophagy induced by serum deprivation [66]. The expression of FADD and concurrent activation of caspase-8 may inhibit hyperautophagy and necroptotic death and favor apoptotic death [149]. Although the crosstalk between apoptosis and autophagy may vary depending on the cell types and stimulus, a better understanding of FADD-mediated regulation of both pathways could be valuable for designing a common strategy to regulate both processes (Figure 5).

### 4.4. Role of FADD in Inflammation and Inflammatory Signaling Pathway

Chronic or persistent inflammation plays a crucial role in the progression of various diseases, including cancer. Cancer cells secure their high proliferative capacity by evading programmed cell death mechanisms through the activation of proto-oncogenes, the NF-κB signal transduction pathway, and the inflammasome. The dysregulation of apoptosis and inflammation represents a characteristic feature of cancer cells and is widely recognized as a hallmark of cancer [150]. In numerous types of cancer, the activation of NF-κB signal transduction pathways and the involvement of immune cells lead to the release of pro-inflammatory cytokines and the activation of pro-oncogenes. These processes modulate cell death regulatory genes and proteins, thereby creating a tumor microenvironment conducive to tumor growth and malignancy. The role of FADD is closely linked to immune defense mechanisms against bacterial and viral infections. FADD has been identified as a significant modulator of innate immunity and inflammatory responses, in addition to its role in regulating apoptosis and non-apoptotic functions [151,152]. Recent studies have unveiled the involvement of FADD in the regulation of inflammation via NLPR-3 inflammasome and FADDosome signaling pathways [153,154]. The formation of the multimolecular complex known as the FADDosome, which consists of caspase-8-FADD-RIPK1, is associated with the NF-κB depended expression of proinflammatory cytokines and chemokines induced by TRAIL (TNF-related apoptosis-inducing ligand) to trigger inflammation and downstream signaling [153]. In A549 cells, the removal of *FADD* or *caspase-8* failed to activate NF-κB and the production of pro-inflammatory cytokines in response to TRAIL. Additionally, the injection of *FADD* knockout A549 cells in mice resulted in the development of lung tumors, highlighting the role of the TRAIL-FADD-NF-κB signaling axis in cytokine and tumor regulation [155]. Essentially, the binding of TLR4/IL-1R triggers the interaction of the adaptor protein MyD88 (myeloid differentiation primary response 88) and downstream IL-1-receptor-associated kinase (IRAK) through DD interactions, leading to the activation of NF-κB signaling and the expression of pro-inflammatory cytokines (IL-6, IL-1β, and TNF) [156,157]. However, FADD can also compete with IRAK for DD interactions and interact with MyD88, thus impairing NF-κB activation and downstream pro-inflammatory signaling [152,158]. Moreover, the loss of FADD propel MyD88-IRAK1 interaction, suggesting that FADD balances the IRAK1 binding to MyD88 in response to the TLR4 activation [159,160]. Depletion of FADD in myeloid cells induces RIP3- and MyD88-dependent systemic inflammation [151]. Another interesting finding suggests that FADD may differentially regulates Fas signaling of apoptosis and inflammation depending on the cell types and stimulation [161].

The NLRP3 (NOD-like receptor family, pyrin domain containing 3) inflammasome is activated in response to a variety of endogenous and exogenous stimuli or danger signals, including ATP, microbial toxins, ROS, ion flux, mitochondrial dysfunction, and lysosomal damage [162]. The NLRP3 inflammasome, a crucial element of innate immunity, plays a pivotal role in the host’s immune responses to pathogens. The assembly of the NLRP3 inflammasome, which includes NLRP family receptors, the adaptor protein ASC, and inflammatory caspase-1, is accountable for the processing and activation of the cytokines IL-1β and IL-18 [162]. In macrophages, the NF-κB-TRIF-MyD88 signaling axis stimulated by LPS primes the assembly of the NLRP3 inflammasome and the expression of pro-IL-1β and pro-IL-18, leading to the maturation of these cytokines [163]. Additionally, caspase-8–FADD–RIPK1 complex has been reported to activate the NLRP3 inflammasome in human monocytic cell lines in response to LPS stimulation, independent of their apoptotic functions [164]. The genetic ablation of *caspase-8* or *Fadd* in murine macrophages impairs both the transcriptional priming and activation of the NLRP3 inflammasome [165]. The activation of the NLRP3 inflammasome leads to the cleavage of gasdermin D (GSDMD) into N-terminus GSDMD (N-GSDMD), releasing large amounts of inflammatory cytokines and inducing inflammatory cell death known as pyroptosis [166].

Pyroptosis plays a crucial role in the eradication of diverse bacterial and viral infections through a cell death mechanism governed by GSDMD and MLKL equilibrium. The anti-apoptotic protein Bcl-2 modulates pyroptosis by controlling the cleavage of GSDMD [28]. MLKL and RIP3 are pivotal signaling proteins accountable for necroptosis and for orchestrating inflammatory mediators associated with diverse chronic inflammatory conditions [167]. RIPK3 ablation effectively prevents embryonic lethality in mice lacking FADD or caspase-8. Additionally, mice deficient in MLKL and FADD demonstrate lymphoproliferative disease and activation of the NLRP3 inflammasome, with the expression of FADD mitigating the disease [168]. A particularly intriguing recent discovery has unveiled that FADD plays a role in the regulation of gut homeostasis and inflammation by governing cell death in intestinal epithelial cells (IECs) through caspase-8, MLKL, and GSDMD mechanisms. Deficiencies in FADD or caspase-8 have been found to facilitate colitis development via MLKL-mediated necroptosis in epithelial cells [169]. Notably, an investigation has shown that the activation of the NLRP3 inflammasome in human monocytes/macrophages induces the secretion of FADD through microvesicle shedding without an increase in IL-1β release and pyroptosis [4]. Activation of the inflammasome is associated with the regulation of the mammalian target of rapamycin (mTOR), a serine/threonine kinase, with a significant role in modulating levels of both pro- and anti-inflammatory cytokines [29]. Furthermore, NLRP3 inflammasome-mediated pyroptosis acts as a protective mechanism against viral infections, such as SARS-CoV-2, preventing a productive viral cycle [170]. Another study demonstrated that the co-treatment of TNF-α and IFN-γ induces the JAK/STAT1/IRF1 axis, leading to caspase-8/FADD-mediated PANoptosis (combination of pyroptosis, apoptosis, and necroptosis) in murine bone marrow-derived macrophages (BMDM. Blocking TNF-α and IFN-γ protected mice from mortality during SARS-CoV-2 infection [171], highlighting the significance of these findings in developing therapies targeting cytokine storm-induced mortality in COVID-19 [172,173,174]. Additionally, the proper antimicrobial responses of the innate immune cells and IECs, such as macrophages and Paneth cells, play crucial roles in regulating gut immune homeostasis [175,176]. When these responses fail to maintain gut homeostasis, chronic inflammation causes illnesses such as inflammatory bowel disease (IBD) [177,178]. In mouse models with IEC-specific deficiencies, caspase-8 deficiency (*Casp8*^fl/fl^ × Vil1-cre, *Casp8*^IEC-KO^) has been reported to develop ileitis [179], as well as impaired mucosal barrier function and bacterial clearance at the epithelial interface, which leads to colitis [180]. Moreover, mice with IEC-specific FADD deficiency (*FADD*^IEC-KO^) spontaneously developed epithelial cell necrosis with loss of Paneth cells and erosive colitis [18]. Interestingly, the regulatory connection of FADD was demonstrated with rheumatoid arthritis, and it has been suggested that FADD negatively regulates IL-1R/TLR4 signaling [181]. Thus, FADD may act as an important scaffold to regulate the inflammatory response, apart from playing a key role in cell death pathways to determine cell fate (Figure 5).

## 5. Therapeutic Potential of FADD 

Cytosolic expression of the adaptor protein FADD is crucial for death receptor-mediated pathways and may serve as a promising therapeutic target in various disease conditions, such as malignancy, autoimmunity, and inflammation. Previous studies have demonstrated that altering FADD expression in T cells impairs resistance against Fas ligand (FasL)-induced apoptosis and promotes cell proliferation [182]. Additionally, FADD-deficient mice (FADD^−/−^) develop thymic lymphoma as they age [183]. Dysregulated expression of FADD could serve as a prominent tumor biomarker and prognostic factor for developing appropriate treatment strategies [1]. We have previously demonstrated that FADD can effectively target the anti-apoptotic protein cFLIP and the pro-inflammatory NF-κB pathway in various tumor cell types [35,36,118,184,185], highlighting the potential of FADD as a therapeutic candidate. Previous approaches have successfully delivered a fusion of the *FADD* gene with the human telomerase reverse transcriptase (hTERT) promoter, resulting in significant apoptosis induction in glioma cells [186]. Advancements in cancer therapy provide opportunities for manipulating the expression of apoptotic genes, which can be utilized to regulate a wide spectrum of pathways [187]. Previous studies have demonstrated that adenovirus or retrovirus-mediated transfer of the FADD gene induces apoptosis in glioma cells [188]. Shinoura et al. showed that adenoviral delivery of FADD adenovirus (Adeno-FADD) potentially induced apoptosis in Fas ligand-resistant U251 glioma cells, suggesting that FADD could be a therapeutic modality for treating gliomas [189]. FADD has the potential to be exploited for the treatment of non-cancer diseases, such as rheumatoid arthritis (RA). Kobayashi et al. utilized adenoviral-mediated delivery of the *FADD* gene and observed the elimination of human rheumatoid synoviocytes engrafted into severe combined immunodeficiency mice. This suggests that FADD gene transfer might be effective in the treatment of RA [190]. Ho et al. demonstrated that viral vector-mediated delivery of FasL and FADD effectively induced cell death in human glioma cells cultured from biopsy samples. Combined therapies of both genes, in the presence of temozolomide, significantly improved the survival of mice bearing high-grade gliomas [191]. Furthermore, novel approaches involving the design of cell-penetrating peptides (CPPs) for the direct delivery of proteins into the cytoplasm of cells improve the prospects of developing cures for several incurable diseases [192,193]. Our previous research has shown that TAT peptide-conjugated FADD protein is successfully delivered to cancer cells through the caveolar pathway of endocytosis [194] and induces apoptosis signaling, simultaneously targeting pro-tumorigenic and pro-inflammatory NF-κB signaling [195]. Collectively, targeted delivery of genes or proteins could be effective in combination with conventional chemo- or radiotherapy for cancer treatment.

## 6. Future Perspective

In this review, we have provided a comprehensive description of the dynamic role of FADD in the regulation of cell death and inflammatory pathways. We have also elucidated the molecular mechanisms through which FADD modulates downstream signaling, including NF-κB activation, RIPoptosome assembly, NLRP3 inflammasome signaling, and pyroptosis. Further advancements in this field will necessitate a more profound mechanistic understanding of FADD-mediated regulation of apoptotic and inflammatory pathways. Thus, understanding the cellular dynamics of FADD in the regulation of programmed cell death signaling and inflammation may shed light on the potential of this protein for the development of novel treatment strategies.

## Figures and Tables

**Figure 1 ijms-25-03228-f001:**
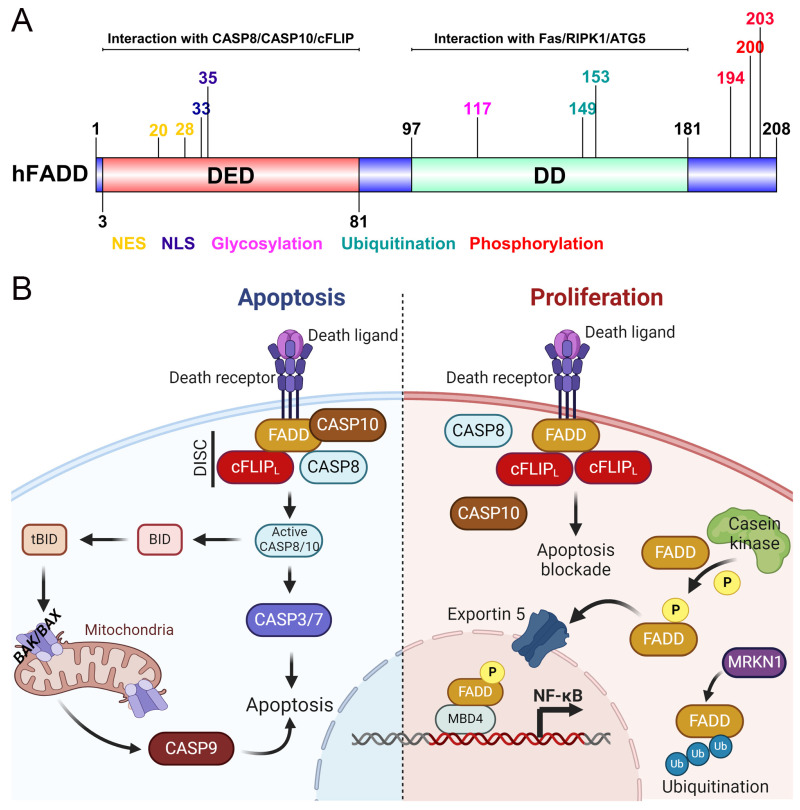
Structure and signaling of FADD. (**A**) Human FADD (hFADD) contains two domains: an N-terminal Death Domain (DD) and a C-terminal Death Effector Domain (DED). DD of FADD interacts with Fas or Atg5 or RIPK1, and DED of FADD interacts with the DED of procaspase-8/10 or cFLIP proteins. (**B**) Sufficient availability of cytosolic FADD protein assembles DISC. However, phosphorylation or ubiquitination of FADD abrogates DISC assembly and leads to cell proliferation. NES, nuclear export signal; NLS, nuclear localization signal. Created with BioRender.com accessed on 19 January 2024.

**Figure 2 ijms-25-03228-f002:**
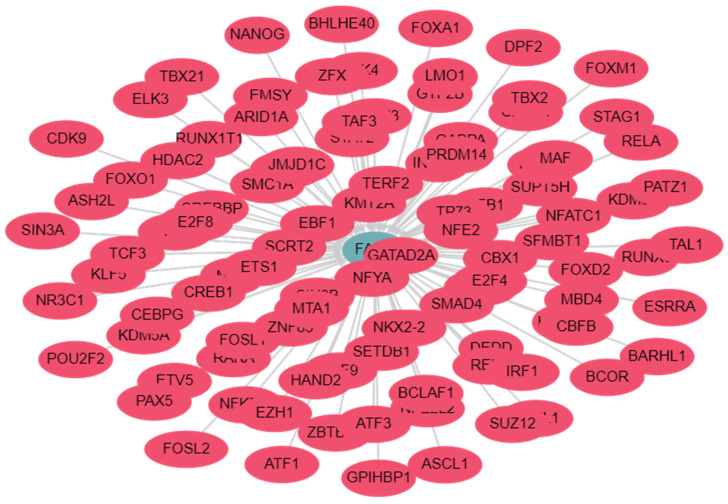
Interacting network of transcription factors (TFs) regulating *FADD*. Interactive network visualization of FADD and regulatory TFs. Total of 411 TFs experimentally determined to regulate FADD promoter region. Datasets obtained from TFlink. Please refer to Appendix A for in-depth analysis [75].

**Figure 3 ijms-25-03228-f003:**
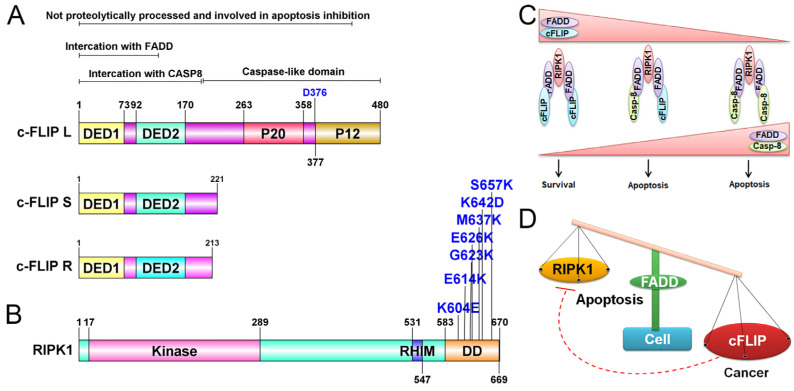
Structure and signaling of cFLIP in ripoptosome assembly. (**A**) Isoform of cFLIP protein. cFLIP_L_ has a catalytically inactive caspase-8-like domain. cFLIPs and cFLIP_R_ have no catalytic domains. All the isoforms contain two death effector domains (DEDs) and interact with FADD and caspase-8/10. (**B**) RIPK1 protein domains and mutation sites (blue) in DD are critical to interact with the DD of FADD. (**C**) Molecular complex of FADD with cFLIP or caspase-8 in the ripoptosome assembly. The levels of FADD are a critical determinant of RIPK1-mediated apoptosis. (**D**) Cellular balance of FADD-cFLIP-RIPK1 in the regulation of apoptosis and cancer.

**Figure 4 ijms-25-03228-f004:**
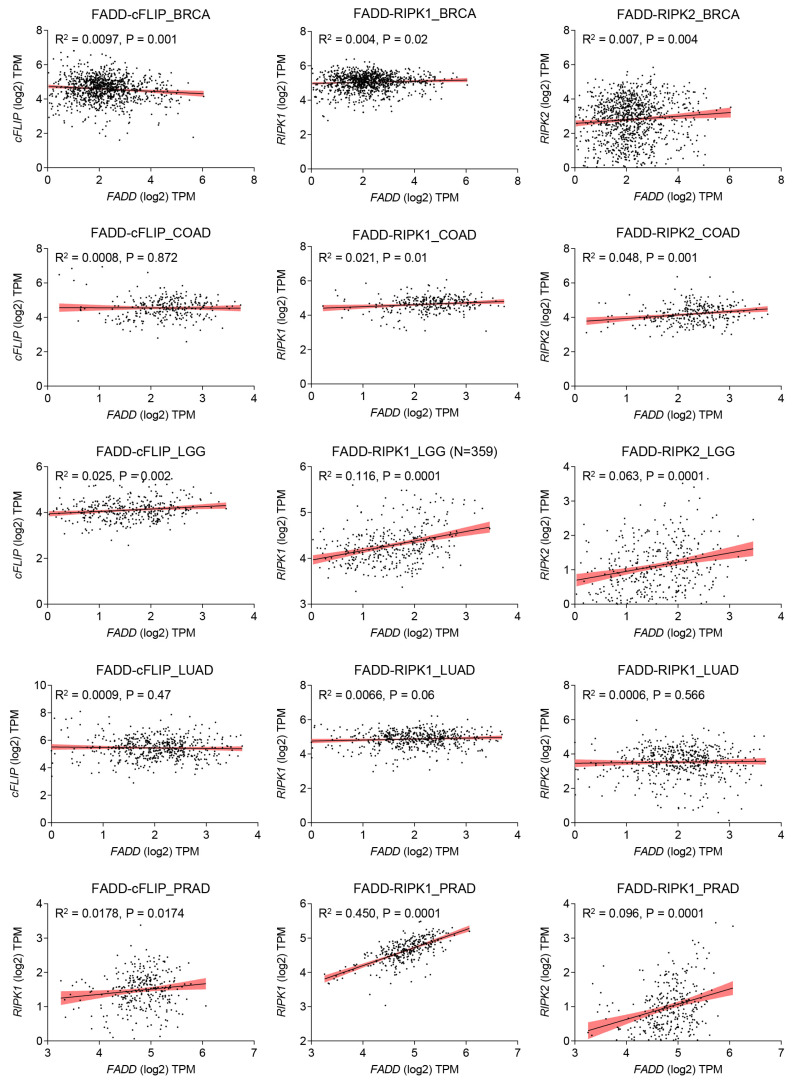
mRNA expression correlation analysis of *FADD* with *cFLIP*, *RIPK1*, and *RIPK2* across different cancer types. The analysis focuses on mRNA expression correlation in various cancer types, including Breast Invasive Carcinoma (BRCA, n = 1063), Colon Adenocarcinoma (COAD, n = 302), Brain Lower Grade Carcinoma (LGG, n = 359), Lung Adenocarcinoma (LUAD, n = 530), and Pancreatic Adenocarcinoma (PRAD, n = 317). TPM, transcripts per million. The datasets used for this study were obtained from oncoDB [98].

**Figure 5 ijms-25-03228-f005:**
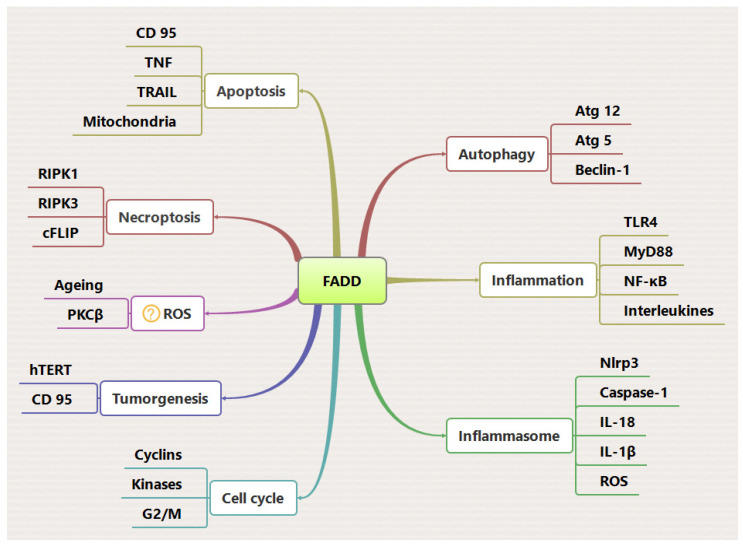
FADD as a master regulator of cell death and inflammatory pathways. Cytosolic expression of FADD is crucial for the regulation of various intracellular pathways. The FADD protein either directly interacts with key pathway regulators (through DD interaction) or induces some pathways through post-translational modifications.

## Data Availability

Not applicable.

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
