# Peer review of "Cellular Dynamics of Fas-Associated Death Domain in the Regulation of Cancer and Inflammation"

_ijms, 2024, doi:10.3390/ijms25063228_

Round 1

Reviewer 1 Report

Comments and Suggestions for Authors

This review by Ranjan and Pathak addresses the different roles played by the Fas Associated Death Domain (FADD) adapter protein FADD. In fact, in addition to papoptotic signaling IFADD is also involved in signaling pathways related to autophagy, cell proliferation and aging and other forms of death. Furthermore, FADD has emerged as a regulator of inflammation contributing to the immune response and cellular homeostasis. In addition to expression, the subcellular localization of FADD also plays a crucial role in determining its different functions. This review would like to provide a complete picture of the functions of FAPP and the biochemical and molecular pathways underlying these functions.

General comments

The topic is potentially interesting and FADD is a molecule with multiple activities that could deserve dedicated review paper. However, this manuscript does not achieve its objectives.

It is written very roughly, with many errors and typos. It often delves into molecular mechanisms, losing sight of the more general aspect which could be the most interesting for the reader.

Specific points

1.     In Figure 1 the authors report the changes in FADD expression in different tumor types compared to their respective healthy tissues. What we notice is that in some it increases and in others it decreases. Meanwhile, the statistical significance between healthy tissue and the corresponding tumor tissue is not reported. Furthermore, if the expression of FADD in tumors is so variable, what is the message of the figure? Why would FADD be crucial in the oncology field? Perhaps the tumors that express it most have particularly aggressive characteristics or, on the contrary, less aggressive ones? The meaning of this figure in the paper is completely obscure to me.

2.     2. Paragraph 2.2 "Previous study demonstrated that the nuclear localization signal (NLS) and nuclear export signals (NES) in the DED facilitate the nucleo-cytoplasmic shuttling of FADD [8]." This sentence is not preceded or followed by any reflection. Why would this aspect be important? A review is not a mere list of letterature data but a critical review based also on the experience of the authors who are assumed to work in the specific field.

3.     Paragraph 2.2 "Importantly, further exploration of the unidentified function of nuclear FADD will contribute to our understanding of its nuclear significance apart from its well-established role in of apoptosis signaling transduction". The apoptotic signal in which FADD participates is not linked to its interaction with death receptors that facilitates molecular interactions between death receptors (DRs) and apical procaspase-8 and -10 to form a multimeric death inducing signaling complex (DISC) for the initiation of apoptosis signaling as mentioned by the authors in the introduction?

4.     Paragraph 2.2 "Importantly, the nucleo-cytoplasmic shuttling protein exportin-5 interacts with FADD to facilitate the import of phosphorylated FADD (pFADD) into the nucleus, and a mutation at Serine 194 (Ser194) disrupts FADD-exportin 5 interactions [10,55]." The mutation in which, FADD or exportin?

5.     We continually switch from the role of nuclear to cytoplasmic FADD. Maybe they should separate from each other.

6.     paragraph 2.2 "Importantly, all three isoforms of cFLIP are believed to competitively inhibit procaspase-8 recruitment to the DISC through their DEDs [62,63]". to dedicate an entire chapter of the review to the interaction between FLIP and FADD the authors should take into consideration its interaction with all three isoforms of FLIP and analyze their effect on apoptosis.

7.     Figure 3. Once again, the expression of FADD versus other proteins such as cFLIP, RIPK1, and RIPK2 is reported in a series of human tumors. What is the meaning of this figure. What emerges from this analysis?

8.     Paragraph 3. "Panner et al. demonstrated PTEN-Akt-AIP4-mediated ubiquitination of FLIPS and sensitivity toTRAIL [77]. Furthermore, lysine 167 (K167) residue of cFLIPL serves as a novel ubiquitination site for ROS-dependent degradation [78] , and the DNA repair protein Ku70 interacts with cFLIP and protects it from polyubiquitination and proteasomal degradation [79]."These sentences contain many messages that should, however, be made explicit. Otherwise, remove from manuscript.

9.     Section 3.2 "Earlier structure-based mutagenesis studies revealed that, RIP1 DD point mutations K604E, E614K, G623K, E626K, M637K, K642D, and S657K disrupt the stability of the complex with FADD DD [87]." In this case, a scheme should be drawn up and the consequences for the cell of this reduced stability of interaction between FADD and RIP1 should also be specified.

10.  Paragraph 4.1.TNF-α exerts its biological effects through cell surface TNF receptors (TNFRs), which consist of a cytoplasmic death domain (DD) approximately 80 amino acids in length. This domain is responsible for recruiting downstream components of the death machinery [68].” TNFRs are made up of several domains, one of which, intracytoplasmic, recruits components of the death machinery. For this to happen the extracellular domain must interact with the ligand (or be constitutively activated).

11.  Paragraph 4.1. "Zhou et al. showed that pharmacological targeting of IAPs suppressed NF-κB activation and induced FADD-dependent apoptosis in multiple myeloma (MM) cells, highlighting the significant functional contribution of FADD [104]." What does this phrase mean? Comment too general. Explain.

12.  Section 4.1 "The existence of these two opposing signaling complexes may explain the lack of response to TNF-α observed in many cells expressing TNF receptors." Why? In cells that have the TNF receptor I would expect a response mediated by complex I or complex II, but not a lack of response to TNF. Can both be activated at the same time?

13.  Figure 4: Very generic. The specific role of FADD is not reported nor what the direct and indirect interactions are. Furthermore, a similar figure should be made describing the role of intranuclear FADD

14.  Paraphrase 4.2 "Thus, FADD-mediated regulation of necroptosis signaling could provide an opportunity to define the fate of cells in pathological consequences (Figure 4)." By pathological consequences do the authors mean pathological conditions? It would be useful to have a schematic figure illustrating in which pathological conditions a role of FADD has been established (increased, decreased, changed, etc.).

Author Response

Response to Reviewer 1

  1. In Figure 1 the authors report the changes in FADD expression in different tumor types compared to their respective healthy tissues. What we notice is that in some it increases and in others it decreases. Meanwhile, the statistical significance between healthy tissue and the corresponding tumor tissue is not reported. Furthermore, if the expression of FADD in tumors is so variable, what is the message of the figure? Why would FADD be crucial in the oncology field? Perhaps the tumors that express it most have particularly aggressive characteristics or, on the contrary, less aggressive ones? The meaning of this figure in the paper is completely obscure to me.

First of all, thank you so much for considering our manuscript potentially interesting and providing valuable comments to revise this manuscript. We have addressed the heterogenous expression of FADD across tumor types, mentioned on pages 6-7 line 268-278 (highlighted in red). However, based on the valuable comments from both the reviewers and after careful consideration we have removed Figure 1 datasets and added structure and signaling of FADD, mentioned on page 4, line 166 (highlighted in red).

  1. Paragraph 2.2 "Previous study demonstrated that the nuclear localization signal (NLS) and nuclear export signals (NES) in the DED facilitate the nucleo-cytoplasmic shuttling of FADD [8]." This sentence is not preceded or followed by any reflection. Why would this aspect be important? A review is not a mere list of literature data but a critical review based also on the experience of the authors who are assumed to work in the specific field.

Thank you for this valuable comment, now we have expanded paragraph 2.2 and added in-depth understanding of the nuclear-cytoplasmic role of FADD, mentioned on page 6, line 228-249 (highlighted in red). We have also Figure 1B to show the nuclear localization of FADD and signaling mechanisms.

  1. Paragraph 2.2 "Importantly, further exploration of the unidentified function of nuclear FADD will contribute to our understanding of its nuclear significance apart from its well-established role in of apoptosis signaling transduction". The apoptotic signal in which FADD participates is not linked to its interaction with death receptors that facilitates molecular interactions between death receptors (DRs) and apical procaspase-8 and -10 to form a multimeric death inducing signaling complex (DISC) for the initiation of apoptosis signaling as mentioned by the authors in the introduction?

Thank you for highlighting this, now we have rephrased the significance of FADD in cytosolic DISC formation and highlighting nuclear localization of FADD in impeding DRs induced apoptotic death. Please refer page 5, line 175-196 and line 212-219 (highlighted in red).

  1. Paragraph 2.2 "Importantly, the nucleo-cytoplasmic shuttling protein exportin-5 interacts with FADD to facilitate the import of phosphorylated FADD (pFADD) into the nucleus, and a mutation at Serine 194 (Ser194) disrupts FADD-exportin 5 interactions [10,55]." The mutation in which, FADD or exportin?

Thank you for highlighting this error, now we have rephrased the sentence. Please refer page 6, line 233-234. (highlighted in red).

  1. We continually switch from the role of nuclear to cytoplasmic FADD. Maybe they should separate from each other.

Thank you for your suggestion, now we have elaborated the nuclear and cytoplasmic FADD roles and added in-depth contents to highlight the pleiotropic role of FADD in the regulation of apoptosis and cell proliferation. Please refer pages 5-6, section 2.2 (changes highlighted in red). Also, we have added a separate Figure 1B for this mechanism.

  1. paragraph 2.2 "Importantly, all three isoforms of cFLIP are believed to competitively inhibit procaspase-8 recruitment to the DISC through their DEDs [62,63]". to dedicate an entire chapter of the review to the interaction between FLIP and FADD the authors should take into consideration its interaction with all three isoforms of FLIP and analyze their effect on apoptosis.

Now we have added structures of cFLIP large, cFLIP short and cFLIP regulator isoform as mentioned on page 8, Figure 3A (highlighted in red). Also, in the text highlighting inhibitory role of cFLIP isoforms in blockingbFADD-caspase-8/10 apoptosis signaling. Please refer page 8, line 326-331 (highlighted in red).

  1. Figure 3. Once again, the expression of FADD versus other proteins such as cFLIP, RIPK1, and RIPK2 is reported in a series of human tumors. What is the meaning of this figure. What emerges from this analysis?

Thanks for your critical analysis, Figure 4 (earlier Figure 3) signifies the regression relationship between the expression of FADD and cFLIP or RIP kinases proteins across different tumor types. No significant regression suggests incoherent expression of FADD and cFLIP or RIP kinases proteins in the specific tumor type and may have independent effect on Ripoptosome assembly or Necroptosis, respectively. Please refer page 11 for Figure 4.

  1. Paragraph 3. "Panner et al. demonstrated PTEN-Akt-AIP4-mediated ubiquitination of FLIPS and sensitivity toTRAIL [77]. Furthermore, lysine 167 (K167) residue of cFLIPL serves as a novel ubiquitination site for ROS-dependent degradation [78] , and the DNA repair protein Ku70 interacts with cFLIP and protects it from polyubiquitination and proteasomal degradation [79]."These sentences contain many messages that should, however, be made explicit. Otherwise, remove from manuscript.

Now we have rephrased the text, please refer page 8, line 352-358 (highlighted in red).

  1. Section 3.2 "Earlier structure-based mutagenesis studies revealed that, RIP1 DD point mutations K604E, E614K, G623K, E626K, M637K, K642D, and S657K disrupt the stability of the complex with FADD DD [87]." In this case, a scheme should be drawn up and the consequences for the cell of this reduced stability of interaction between FADD and RIP1 should also be specified.

Thanks for the suggestions, we have added a scheme for the respective mutation and rephrased the text, please refer Figure 3B and page 9, line 379-383 (highlighted in red).

  1. Paragraph 4.1.TNF-α exerts its biological effects through cell surface TNF receptors (TNFRs), which consist of a cytoplasmic death domain (DD) approximately 80 amino acids in length. This domain is responsible for recruiting downstream components of the death machinery [68].” TNFRs are made up of several domains, one of which, intracytoplasmic, recruits components of the death machinery. For this to happen the extracellular domain must interact with the ligand (or be constitutively activated).

Binding of TNFα to TNF receptor induces conformational changes to oligomerize DD containing adaptor proteins and other signaling molecules. Now we have rephrased the text, please refer page 10, line 427-430 (highlighted in red).

  1. Paragraph 4.1. "Zhou et al. showed that pharmacological targeting of IAPs suppressed NF-κB activation and induced FADD-dependent apoptosis in multiple myeloma (MM) cells, highlighting the significant functional contribution of FADD [104]." What does this phrase mean? Comment too general. Explain.

We have rephrased the text for clarity, please refer page 10, line 452-456 (highlighted in red).

  1. Section 4.1 "The existence of these two opposing signaling complexes may explain the lack of response to TNF-α observed in many cells expressing TNF receptors." Why? In cells that have the TNF receptor I would expect a response mediated by complex I or complex II, but not a lack of response to TNF. Can both be activated at the same time?

Thank you for highlighting this and we agree with your comment. Now we have rephrased the sentence. Please refer page 10-11, line 460-464. (highlighted in red).

  1. Figure 4: Very generic. The specific role of FADD is not reported nor what the direct and indirect interactions are. Furthermore, a similar figure should be made describing the role of intranuclear FADD

Now we provided a separate Figure 1B to describe nuclear FADD and discuss their role in the revised text on page 6, line 228-249 (highlighted in red).

  1. Paraphrase 4.2 "Thus, FADD-mediated regulation of necroptosis signaling could provide an opportunity to define the fate of cells in pathological consequences (Figure 4)." By pathological consequences do the authors mean pathological conditions? It would be useful to have a schematic figure illustrating in which pathological conditions a role of FADD has been established (increased, decreased, changed, etc.).

Thank you for highlighting this and to clarify we it should ne pathological conditions. Now we have corrected the text. Please refer page 13, line 510-512. (highlighted in red). Although we appreciate the merit of this suggestion to provide a schematic figure illustrating role of FADD in pathological conditions, instead we have revised section 4.4 and provided in-depth literature support regarding immune related diseases associated with FADD after considering the space limit and scope of our current review.  

Reviewer 2 Report

Comments and Suggestions for Authors

In this manuscript, the authors introduce the structure, localization, and expression of FADD protein, and then the interaction of FADD protein with other molecules in the cell; then the authors introduce the role of FADD in cell death and inflammatory signaling pathways; and finally, the authors introduce the role of FADD in the treatment of cancer. The authors attempt to provide a systematic overview of the role of FADD in regulating cell death and inflammatory responses. Overall, this manuscript is interesting, but I think the authors may have missed some key information.

1.      As shown in Figure 1, the trend of FADD protein expression is different in different cancer tissues compared to normal tissues. FADD protein expression becomes higher in some cancers (e.g. cholangiocarcinoma) and lower in some cancers (e.g. prostate adenocarcinoma). But the authors seem to have focused only on the types of cancers in which FADD expression was downregulated. And there is no mention of whether FADD is regulated differently in these different cancers separately in the following text. However, this is a problem that deserves attention.

2.      I suggest that the authors include a figure showing the localization and structure of the FADD gene and protein to help readers quickly visualize the basic information about this protein.

3.      In the section "Role of FADD in cell death and inflammatory signaling". Regarding FADD and cell death, the authors only mentioned the role of FADD in necroptosis and autophagy. However, there are more ways of cell death than these two, besides the two mentioned by the authors, there is also apoptosis, pyroptosis, ferroptosis, etc. And there are many literatures reporting the correlation between FADD and apoptosis, but the authors did not mention it in the manuscript. I suggest that the authors review more literature to gather more information on the links between different cell death modes and the FADD protein.

4.      In the section "FADD in cancer therapeutic", the authors mention the role of the FADD gene in the treatment of rheumatoid arthritis (ref. 166). However, rheumatoid arthritis is not a cancer and its inclusion in this section does not fit the title.

5.      The authors mention cancer and inflammatory diseases in the title of the manuscript. The authors then devote a section of the manuscript to the role of FADD in the treatment of cancer, but do not see a section describing the treatment of inflammatory diseases. In addition, as FADD expression tends to be different in different cancers, it is suggested that the authors divide these cancers into two categories and describe the mode of action of FADD in cancer types where FADD is up- or down-regulated, as well as the effect on the prognosis of the cancer.

Author Response

Response to reviewer 2

  1. As shown in Figure 1, the trend of FADD protein expression is different in different cancer tissues compared to normal tissues. FADD protein expression becomes higher in some cancers (e.g. cholangiocarcinoma) and lower in some cancers (e.g. prostate adenocarcinoma). But the authors seem to have focused only on the types of cancers in which FADD expression was downregulated. And there is no mention of whether FADD is regulated differently in these different cancers separately in the following text. However, this is a problem that deserves attention.

First of all, thank you so much for considering our manuscript interesting and providing valuable comments to revise this manuscript. We have addressed the heterogenous expression of FADD across tumor types, mentioned on pages 6-7 line 268-278 (highlighted in red). However, based on the valuable comments from both the reviewers and after careful consideration we have removed Figure 1 datasets and added structure and signaling of FADD, mentioned on page 4, line 164 (highlighted in red).

  1. I suggest that the authors include a figure showing the localization and structure of the FADD gene and protein to help readers quickly visualize the basic information about this protein.

Thank you for your suggestion, now we have expanded paragraph 2.2 and added in-depth understanding of the nuclear-cytoplasmic role of FADD, mentioned on page 6, line 175-197 (highlighted in red). We have also Figure 1B to show the nuclear localization of FADD and signaling mechanisms.

  1. In the section "Role of FADD in cell death and inflammatory signaling". Regarding FADD and cell death, the authors only mentioned the role of FADD in necroptosis and autophagy. However, there are more ways of cell death than these two, besides the two mentioned by the authors, there is also apoptosis, pyroptosis, ferroptosis, etc. And there are many literatures reporting the correlation between FADD and apoptosis, but the authors did not mention it in the manuscript. I suggest that the authors review more literature to gather more information on the links between different cell death modes and the FADD protein.

Thank you for highlighting this and we agree with your comment. Now we have reviewed relevant literatures and provided in-depth role of FAD in other forms of cell death. Please refer page 2-3, line 92-113. Page 15, line 596-613, (highlighted in red).

  1. In the section "FADD in cancer therapeutic", the authors mention the role of the FADD gene in the treatment of rheumatoid arthritis (ref. 166). However, rheumatoid arthritis is not a cancer and its inclusion in this section does not fit the title.

Thank you for highlighting this error, now we have rephrased the sentence. Please refer page 6, line 233-234. (highlighted in red). Please refer page 16, line 657-661, (highlighted in red).

  1. The authors mention cancer and inflammatory diseases in the title of the manuscript. The authors then devote a section of the manuscript to the role of FADD in the treatment of cancer, but do not see a section describing the treatment of inflammatory diseases. In addition, as FADD expression tends to be different in different cancers, it is suggested that the authors divide these cancers into two categories and describe the mode of action of FADD in cancer types where FADD is up- or down-regulated, as well as the effect on the prognosis of the cancer.

As correctly asked by the reviewer, and we agree with the reviewer that the major section of this manuscript discussed the role of FADD in the treatment of cancer. In the revised manuscript we have discussed in-depth role of FADD in inflammatory signaling in the context of inflammatory diseases. To maintain clarity and coherence, we have changed the title of the manuscript to Cellular dynamics of FADD in the regulation of Cancer and Inflammation.  We have discussed the heterogeneous nature of FADD in different cancer types, please refer pages 2-3, line 89-113 and pages 6-7, line 268-278 (highlighted in red).  

Round 2

Reviewer 1 Report

Comments and Suggestions for Authors

The authors addressed  all the questions raised. I have no further requests.

Reviewer 2 Report

Comments and Suggestions for Authors

The manuscript has been completely revised.